# Effect of MWCNTs on Wear Behavior of Epoxy Resin for Aircraft Applications

**DOI:** 10.3390/ma13122696

**Published:** 2020-06-12

**Authors:** Mateusz Mucha, Aneta Krzyzak, Ewelina Kosicka, Emerson Coy, Mikołaj Kościński, Tomasz Sterzyński, Michał Sałaciński

**Affiliations:** 1Faculty of Aviation, Military University of Aviation, Dywizjonu 303 35, 08-521 Dęblin, Poland; a.krzyzak@law.mil.pl; 2Faculty of Mechanical Engineering, Lublin University of Technology, Nadbystrzycka 36, 20-618 Lublin, Poland; e.kosicka@pollub.pl; 3NanoBioMedical Centre, Adam Mickiewicz University, Wszechnicy Piastowskiej 3, 61-614 Poznań, Poland; coyeme@amu.edu.pl (E.C.); mikolaj.koscinski@amu.edu.pl (M.K.); 4Department of Physics and Biophysics, Faculty of Food Science and Nutrition, Poznań University of Life Sciences, Wojska Polskiego 38/42, 60-637 Poznań, Poland; 5Polymer Division, Institute of Materials Technology, Poznan University of Technology, Piotrowo 3, 61-138 Poznań, Poland; tomasz.sterzynski@put.poznan.pl; 6Air Force Institute of Technology, Księcia Bolesława 6, 01-494 Warsaw, Poland; michal.salacinski@itwl.pl

**Keywords:** composites, carbon nanotubes, wear, mechanical properties, epoxy resin

## Abstract

The aim of the study is to assess the effect of multi-walled carbon nanotubes (MWCNTs) on the wear behavior of MWCNT-doped epoxy resin. In this study, a laminating resin system designed to meet the standards for motor planes was modified with MWCNTs at mass fractions from 0.0 wt.% to 2.0 wt.%. The properties of the carbon nanotubes were determined in Raman spectroscopy and HR-TEM. An examination of wear behavior was conducted on a linear abraser with a visual inspection on an optical microscope and SEM imaging, mass loss measurement, and evaluation of the wear volume on a profilometer. Moreover, the mechanical properties of MWCNTs/epoxy nanocomposite were evaluated through a tensile test and Shore D hardness test. The study shows that the best wear resistance is achieved for the mass percentage between 0.25 wt.% and 0.5 wt.%. For the same range, the tensile strength reaches the highest values and the hardness the lowest values. Together with surface imaging and a topography analysis, this allowed describing the wear behavior in the friction node and the importance of the properties of the epoxy nanocomposite.

## 1. Introduction

The observed development of science and technology leads to the use in machines and devices of such friction nodes in which the increased maximum load and speed of their movement occur. Due to the need to minimize the wear of tribological pairs [1,2], the possibilities of limiting the effects of friction are being investigated, which in effect is to contribute to the reduction of energy consumption and cooperating surfaces of machinery and equipment [3]. This is of vital importance, especially in the context of sensitive areas of material use, in which safety is taken into account before the economic criteria, which includes e.g., aviation industry [4,5]. Therefore, material engineering is becoming one of the key determinants of technological progress [6].

Currently, various modifiers allow achieving a wide range of properties for new materials [7,8,9,10,11,12]. As an example, studies conducted for aviation over modified polymers are most often carried out in order to improve material strength or electrical conductivity [13,14,15]. The improved electrical conductivity is desirable due to the lightning strike protection (LPS). In this case, it is important to determine the filler concentrations to obtain the best results. For instance in case of conductive carbon black in composites based on low-density polyethylene (LDPE) and poly(ethylene-co-vinyl acetate) (EVA), the percolation threshold was obtained for values higher than 15 wt.% [16].

The influence of MWCNTs on hardness, impact strength, and thermal conductivity of epoxy resin composites is widely discussed [10]. Other tests assessing the structural usefulness of a material include a hardness test [17,18]. Longitudinal and transverse shrinkage as well as plastic and elastic deformations [19] are also important in the context of materials for aviation [20,21]. Another important challenge in all technical applications of epoxy resin based composites is the need to produce materials with strongly reduced flammability [22,23].

Developing nanotechnology allows the use of increasingly sophisticated nanocomposites [24]. The nanofillers used include nanofibers [25,26], graphene flakes [27], and carbon nanotubes [28,29]. When descending to the nanoscale, dispersion of the nanofiller becomes crucial. The problems of inhomogeneous dispersion of nanofillers, agglomeration of MWCNTs, poor interfacial adhesion, etc. are widely discussed [30]. The creation of polymer nanocomposites with a homogeneous distribution of MWCNTs as a nanofiller is a very complex problem as shown in many papers [31,32,33,34,35]. To solve the dispersion problem, many solutions have been suggested, such as the addition of dispersing agents, shear mixing [28], and functionalization [29,36,37]. For example, the study on composites based on polypropylene (PP) presented in [38] suggests that the form of nanofiller dispersion such as the Masterbatch (MB) dilution is suitable for the dispersion of MWCNTs but at the expense of carbon nanotube shortening and, as a consequence, an increase in electrical resistivity. The homogeneity of MWCNT in the composite can be determined by means of the high-resolution scanning electron microscopy (HRSEM) [39].

The addition of nanoparticles to thermosetting matrix materials assists in strengthening the surface, which results in enhancing the tribological behavior of polymers. This is especially true for adhesive wear loading conditions under dry contact [40]. The filler addition greatly enhances the tribological properties of the epoxy resin, by reducing the friction coefficient and the wear rate [29]. As far as wear tests are concerned, a different test equipment is in use: from normalized ball on disc tribotesters to an in-house designed and built rubber wheel/dry sand test equipment [41] or original equipment designed and fabricated by the university [42].

In studying the application of carbon nanotubes to a piston ring and cylinder liner system, it was found that the friction coefficient decreases with an increase in MWCNTs content. That is a result of the fact that, under dry friction, the MWCNTs act as a solid lubricant and form a carbon film covering the contact surfaces [43]. For example, the study on the helical carbon nanotubes (H–CNTs) in the mass fraction range from 0.0 wt.% to 2.0 wt.%, presented in [44], showed that the friction coefficient, as well as the wear rate decrease with increasing the nanofiller content. Moreover, fillers can alter the crosslinking process of the polymeric matrix in comparison with the neat epoxy, in particular reduce the gel time of the resin [45,46].

Tools, such as TEM, are needed to verify the quality of nanocomposites. High-resolution transmission electron microscopy (HR-TEM) provides better insights into the physical behavior of many nanostructured materials [47]. An example of using it for research in which carbon nanotubes were used is presented in [48]. The image is formed by the interference of the diffracted beams with the direct beam. This is called phase contrast. HR-TEM images are obtained when the point resolution of the microscope is sufficiently high and a crystalline sample oriented along a zone axis [49].

Raman spectroscopy is also an important tool in the field of nanotechnology. Raman scattering is a component of Raman spectroscopic techniques and is used to obtain information about the structure and properties of molecules. This information comes from their vibrational frequencies. Raman scattering is a two-photon event. As known, the process focuses on the change in polarizability of a molecule which take account of its vibrational motion [50,51]. In [52], the authors focus on this method and describe its essence in detail. Some current applications of Raman spectroscopy are observed in the fields of biomedical diagnostics [53,54], archaeological science [55], industrial process control, environmental science [56], astrobiology, and materials engineering [57].

The investigated MWCNTs mass fraction range can vary depending on the aim of the research. In order to examine the impact on the electrical conductivity the wider range of 0.0–5.0 wt.% was applied in [10]. As far as mechanical properties are concerned, lower concentrations were investigated: 0.0–1.0 wt.% in [58], 1.0 wt.% in [59], 0.1–0.9 wt.% in [15], 0.0–0.5 wt.% in [42,60,61] (and also in [62] but the paper also cites studies for 5.0 wt.%), 0.3 wt.% in [63] (for of carboxyl functionalized MWCNTs), 0.025–0.2 wt.% in [46] and 0.2 wt.% in [37].

In the previous studies on laminating resin system designed to meet the standards for motor planes, modified with MWCNTs, proved that electrical conductivity of MWCNTs/epoxy nanocomposite increased with higher MWCNTs mass fraction even for a very simplified manufacturing procedure [64]. The percolation threshold was obtained in the range between 5.0 wt.% and 6.0 wt.%. However, such high mass fractions are not justified economically and due to the deterioration of mechanical properties and manufacturing problems (increased resin viscosity), are not recommended for structural components. Moreover, it is desirable that such parts exhibit good mechanical properties, such as high tensile strength and high resistance to wear [65,66]. In order to achieve a slight improvement of electrical properties and simultaneously provide a hope for an improvement of abrasion resistance, mass fractions from 0.0 wt.% to 2.0 wt.% were chosen.

The aim of the study is to assess the effect of MWCNTs content on the wear behavior of MWCNT-doped epoxy resin. The novelty of this article is that a linear abrader was used, which allows the comparison of results with measurements made by other abrasion tests. Secondly, Raman spectroscopy and TEM imaging were performed, which is not common in abrasive wear testing. Moreover, carbon nanotubes concentrations of up to 2.0% were used, which is not common in the studies of mechanical properties of epoxy resin nanocomposites. The presented results can be a reference point for the research on wear behavior, as well as epoxy resin modification in order to improve mechanical properties or electrical conductivity.

## 2. Materials

### 2.1. Raw Materials

The matrix of the MWCNTs/epoxy nanocomposite was made of the MGS L285 laminating resin system approved by German Federal Aviation Authority. L285 is intended for use with glass, carbon and aramid fibers and is characterized by high static and dynamic strength. After heating, it meets the standards for motor aircraft, gliders and powered sailplanes. The laminating resin L285 is a mixture of epoxy resin (number average molecular weight ≤ 700), which is a reaction product of bisphenol-A-(epichlorhydrin) (50 wt.%) and 1,2,3-Propanetriol, glycidyl ethers (50 wt.%). The specification of L285 is as follows: density (25 °C): 1.18–1.23 g/cm^3^, viscosity: 600–900 mPas/25 °C epoxy equivalent: 165–170, epoxy number: 0.59–0.65. 

There are different hardeners dedicated for the L285 resin. For the experiment, H287 hardener was selected as characterized by a long gelling time, which makes it easier to prepare fiber reinforced composites. H285 hardener is a mono-constituent substance based on 2,2’-dimethyl-4,4’-methylenebis(cyclohexylamine) with possible impurity in the form of 1,4-bis(butylamino)anthraquinone. The H 287 properties are as follows: density (25 °C): 0.93–0.96 g/cm^3^, viscosity: 80–100 mPas/25 °C amine number: 450–500.

For the nanofiller, industrial grade multi-walled nanotubes of 90 wt.%, 10 nm OD, manufactured by Bucky USA (Houston, TX, USA), were chosen. The MWCNTs were delivered in the form of powder. Their properties stated by the manufacturer in the specification were as follows: purity: 90 wt.%, outer diameter (OD): 10–30 nm, inner diameter (ID): 5–10 nm, length: 10–30 µm, specific surface area (SSA): >200 m^2^/g, bulk density: 0.06 g/cm^3^, true density: ~2.1 g/cm^3^.

### 2.2. Sample Preparation

A pure powder of MWCNTs was used for HR-TEM as well as for reference in Raman spectroscopy. The MWCNTs/epoxy nanocomposite was prepared by direct nanotube incorporation. The mechanical mixing and the ultrasound exposure were implemented. The concentrations of carbon nanotubes were selected so that after the addition of the hardener prepared mass fractions are 0.0, 0.25, 0.5, 0.75, 1.0, and 2.0 wt.%. After that, the composition of L285 and MWCNTs was mixed with H287 in 100:40 resin to hardener weight ratio as recommended by the manufacturer.

To achieve this goal, mixing with a mechanical laboratory stirrer was carried out at the speed of 300 rpm for 10 min. Then, to remove air bubbles, the mixture was placed in a vacuum chamber. The next step was exposure to working frequency of 40 kHz in the Ultron ultrasonic bath for 10 min. Then the composite was casted in silicone molds (Figure 1b) and between sheets of PCV films. 

Specimens for tensile testing, hardness testing, and abrasion testing were cast in the molds. The silicone forms were designed in accordance with the ISO 527-1 standard (Figure 1a). In order to maintain repeatability in the thickness of the cast samples, the upper surface was processed on a numerically controlled milling machine. This procedure also allowed the examination of the core material bypassing the surface layer, which usually has different properties [67]. The composite compressed between sheets of PVC films allowed obtaining flat samples of the material. They were prepared for Raman spectroscopy and for bright field observation in the optical microscope. 

## 3. Methods

### 3.1. High-Resolution Transmission Electron Microscopy (HR-TEM)

In order to prove that properties of carbon nanotubes are at least as good as those provided by the manufacturer and in order to determine a contaminant and a general composition of the structures, high-resolution transmission electron microscopy (HR-TEM) was performed. HR-TEM was collected on a JEOL (Tokyo, Japan) ARM-200F microscope working at 80 kV equipped with an energy dispersive X-ray (EDX) detector. The samples were prepared by sonicating a solution of MWCNTs powder in pure ethanol. The sonicated product was then drop casted on commercially available Lacey Carbon Cu grids and dried overnight in a vacuum dissector connected to a membrane vacuum pump (<2 mBar).

### 3.2. Raman Spectroscopy

Raman spectra were obtained using an in Via Ranishaw Raman Microscopy system (Ranishaw, Old Town, Wotton-under-Edge, UK) with a 633 nm He/Ne laser (0.75 mW laser power, Stage I) and 1800 g/mm grating. The laser light was focused on the sample with a 20×/0.75 microscope objective (LEICA, Wetzlar, Germany) All Raman spectra were obtained from 450 to 4000 cm^−1^ using 20 s acquisition time. All spectra were corrected by using the WiRETM 3.3 software attached to the instrument. The measurement of peak positions was performed by using Lorentz profile at OriginPro 8.3 software (Northampton, MA, USA).

### 3.3. Tensile Testing

The uniaxial tensile testing was carried out on a Zwick Roell (Ulm, Germany) Z5.0 testing machine, operating with the test speed of 2 mm/min and at normal ambient conditions in accordance with ISO 527-1 standard. 10 samples were tested for each mass fraction of MWCNTs. Based on the results, mean values and standard errors were determined.

### 3.4. Hardness Tests 

Shore D hardness was determined according to ISO 868 standard (Bareiss Shore/IRHD Digi Test II, FRT GmbH, Bergisch Gladbach, Germany). 10 samples were tested for each mass fraction of MWCNTs.

### 3.5. Abrasion Tests

The examination of wear behavior was conducted on TABER Linear Abraser Model 5750 (North Tonawanda, NY, USA) (Figure 2). 

The conditions during the abrasion test were as follows: stroke length: 50.8 mm, speed: 60 cycles per minute, maximum velocity: 159.52 mm/s, total load: 1850 ± 1 g, abradant model: H-18 (vitrified, medium), abradant diameter: 6.35 mm, maximum number of cycles for one sample: 1000, wear intervals: 200 cycles. Five samples were tested for each mass fraction of MWCNTs.

After each 200-cycle interval, mass loss and topography of the sample were measured. The mass measurements were taken on a precision laboratory weight. The evaluation of the topography was performed on the MicroProf 100 surface metrology tool (FRT GmbH, Bergisch Gladbach, Germany) with CWL 600 µm sensor (FRT GmbH, Bergisch Gladbach, Germany). Measuring characteristics were as follows: measuring range z: 600 µm, resolution (lateral): 2 µm, resolution (vertical): 6 nm. Each time the same part of the specimen surface area was mapped: 9 mm × 4.5 mm rectangle with a long side perpendicular to the abrasion direction. 3D data and profiles were analyzed using FRT Mark III software (FRT GmbH, Bergisch Gladbach, Germany). 

### 3.6. Surface Imaging

Visual inspection was carried out on the Olympus (Tokyo, Japan) BX53M optical microscope with 5× magnification. Bright field method was applied to thin films of composite in order to check whether the obtained image had unfavorable microscale agglomerations or if a gradient arrangement of MWCNTs occurs. The dark field method was applied to casted samples before and during the abrasion test. The application of this method allowed highlighting surface irregularities occurring on the sample. The SEM images were collected on Hitachi (Tokyo, Japan) tabletop microscope model TM3030 plus with accelerating voltage 15 kV and 500× magnification.

## 4. Results

### 4.1. HR-TEM

TEM imaging shows MWCNTs outer diameter of 22.3 ± 3.5 nm. The length is variable and difficult to estimate since the tubes are found in bundles. The van der Waals distance between concentrically walls is estimated as 0.289 nm, thus an average of 20 layers is present in the MWCNTs. On the other hand, surface features (marked by yellow arrows), show partial oxidation and damage to the carbon nanotubes. Finally, metallic particles, residual products from the synthesis process, are present and embedded in the carbon nanotubes central sections (Figure 3).

EDX mapping shows relative oxidation of the MWCNTs surfaces and the FeO_x_ structure of the central particles. In the oxidized carbon nanotubes, oxygen groups are covalently attached to nanotubes [68]. These are: carboxyl (–COOH), hydroxyl (–OH), epoxy (C–O–C), and carbonyl (C=O).

Traces of nickel and chromium (in the range of 0.5 wt.%) were also detected (Figure 4). The origin of this contamination is connected with the fabrication method of MWCNTs and the values are typical for chemical vapor deposition (CVD) [69], and therefore it was considered that they did not affect the results of other measurements. Large scale collections allowed determining a relative 95 wt.% purity of the MWCNTs with a small concentration of iron and oxygen.

### 4.2. Raman Spectroscopy

The Raman spectra of 0.25, 0.5, 0.75, 1.0, and 2.0 wt.% MWCNTs content in epoxy matrix and pure reference MWCNTs are presented in Figure 5. 

The characteristic modes: D (double-resonance mode), G (tangential stretching mode), and 2D (two phonon process) at ~1338 cm^−1^, ~1586 cm^−1^, and ~2665 cm^−1^ respectively, confirms the presence of carbon nanotubes in the samples. The D-band is indicative of structural disorder due to disruption of sp^2^ C–C bonds, whereas the G-band results come from the tangential vibration of graphitic carbon atoms. All characteristic modes were observed from 0.75 wt.% MWCNTs mass fraction, for lower MWCNTs mass percentage the maxima are visible, although in a fuzzy form, signifying a weaker Raman reflection. Shifting of MWCNTs peak positions was not observed.

### 4.3. Tensile Testing

Stress–strain curves for all investigated carbon nanotube mass fractions are presented in Figure 6. 

For each concentration of MWCNTs, namely 0.0, 0.25, 0.5, 0.75, 1.0, and 2.0 wt.%, a separate graph is presented showing stress–strain curves for all tested samples. The shapes of the stress–strain curves, indicate the brittle fracture of the composites at break, and in this respect all the presented characteristics are similar. As it may be seen form the graphs, a small addition of carbon nanotubes of only 0.25 wt.% causes almost a two-fold increase in tensile strength and a similar increase in relative deformation.

The average ultimate tensile strength σ_u_ increases for the MWCNTs mass fraction 0.25 wt.% compared to the neat resin by 69% reaching the value of 45.7 MPa and then decreases for higher mass percentages (Figure 7a). Initially increasing and then falling σ_u_ curve in the concentration range from 0.0 to 0.5 wt.% was also observed in [61] for the polyester matrix samples. Initial growth may be due to arresting and delaying the crack growth effect of MWCNTs, as suggested in [63]. There is a 4th degree polynomial approximation for the eye guiding effect. The interpolated value for 0.1 wt.% is 36% higher than the value for the pure resin, which is only 4% greater than the increase given in [42] but at the same time twice the increase given in [60] for pristine CNTs. It can be noticed that the curve has another extreme a between 1.0 and 2.0 wt.% and begins to grow again, although the whiskers presenting a standard deviation suggest that this rise is of negligible importance.

The curve approximating the change of the nominal strain at break ε_m_ as a function of carbon nanotubes mass fraction has the same shape as σ_u_ (Figure 7b). The maximum average value of 2.65% was calculated for MWCNTs 0.25 wt.% which is a 126.5% increase over the resin without a filler. Initially increasing and then falling ε_m_ curve in the mass fraction range from 0.0 to 0.5 wt.% was also observed in [42].

Calculated mean values of Young’s modulus E_t_ with corresponding standard errors are as follows: 2209 ± 56 MPa for 0.0 wt.%, 1876 ± 27 MPa for 0.25 wt.%, 1728 ± 33 MPa for 0.5 wt.%, 1824 ± 24 MPa for 0.75 wt.%, 1760 ± 31 MPa for 1.0 wt.%, and 1804 ± 38 MPa for 2.0 wt.% MWCNTs. All obtained values are higher than results given in [60], in particular E_t_ of pure resin is 74% higher, probably due to the use of different Bisphenol-A epoxy resin, namely ED-22. The obtained results mean a significant decrease for a concentration of 0.0 to 0.5 wt.%, an increase for the range of 0.5 to 0.75 wt.% and insignificant changes for MWCNTs mass fraction above 0.75 wt.%. E_t_ decrease in the concentration range from 0.0 to 0.5 wt.% was also observed in [61] for the polyester matrix samples. The effect of the multi-walled carbon nanotubes concentration on the modulus of elasticity is opposite than on the ultimate tensile strength and the nominal strain at break.

An initial improvement in both mechanical parameters, σ_u_ and ε_m_, is probably due to the effect of an interface between the matrix and the nanofiller. It is also possible that carbon nanotubes transfer the tensile loads through the specimen. A slight deterioration of those properties with the MWCNTs fraction higher than 0.5 wt.% may be a result of a greater concentration of the filler agglomerates. It can be assumed that the change of the toughness is plotted along the similar curve, while the brittleness B (as defined by Brotsow et al. [70]) behaves like inversed toughness (although it is not a rule [71]). The observed increase in toughness may indicate individual nanotubes sliding within the bundles (as noted in [62]). The initial decrease in B (MWCNTs 0.0 wt.% to 0.25 wt.%) could be explained by the improvement of mechanical properties owing to the presence of the filler.

The increase in B (MWCNTs 0.5 wt.% to 1.0 wt.%) could be explained by the impact of the nanofiller concentration on the crosslinking of the polymer. The epoxy resin that is not crosslinked in the whole volume has no tendency to crack. It is probable that the presence of carbon nanotubes catalyzes the crosslinking of the polymer. Another decrease in B may result from increased flexibility of the material due to greater mass fraction of MWCNTs.

### 4.4. Hardness Test

Shore hardness measurement results have been presented in Figure 8. For a mass fraction of MWCNTs 0.25 wt.%, an average hardness value of 76.2 was obtained. This is a decrease of 7% compared to the value for neat resin. As the mass fraction increases, the hardness increases to values above 83, which is higher than the value for the neat resin, although for 2.0 wt.% it decreases again. It is possible that increasing the hardness in the MWCNTs content ranging from 0.25 wt.% to 0.75 wt.% has the same background as σ_u_, as it is discussed on the example of the epoxy-polyamide blend in [72].

### 4.5. Abrasion Tests

Figure 9 shows the wear behavior of a MWCNTs 0.25 wt.% sample presented as an example. The diagram shows the mass loss as a function of the number of wear cycles. Each value on the graph is the difference in sample weight before the first 200-cycle interval and the weight after each interval. The slope of the graph therefore corresponds to the value of a weight loss per cycle. The course of the graph is rather linear, which means that the weight losses during subsequent intervals are similar.

Along with successive intervals of 200 abrasion cycles, the material is gradually penetrated, which is visible on the profiles (Figure 10). As it may be seen on the profiles, volume losses per cycle interval are rather constant and proportional to weight loss

The diagram presenting the average mass loss after 1000 wear cycles, with average values marked after intervals of 200 cycles, is very similar to Shore D hardness results (Figure 8). There is a significant drop to 0.019 g for the mass fractions of MWCNTs 0.25 and 0.5 wt.%. This represents a decrease by 28.8% compared to the neat resin. The obtained result contradicts the result presented in [42] where 38% increase in the weight loss was noted. However, the cited article does not contain a detailed description of the research method, and because of that, there is no possibility to determine the cause of the discrepancy. For MWCNTs 0.75 wt.% an increase to the same value as obtained for 0.0 wt.% was noted and, as in the case of the hardness, a decrease occurred for 2.0 wt.%. The wear rate results correspond well with a general decrease in the range from 0.0 wt.% to 2.0 wt.% presented in [10], however, a greater number of investigated MWCNTs mass fraction values revealed the local minimum between 0.25 wt.% and 0.5 wt.%. The superior value of the specific wear rate at 0.5 wt.% exhibits a good agreement with O. Jacobs et al. [28], as well as with [72]. The correlation between the wear rate and the hardness was also observed by Aruniit et al. for the polymer filled with aluminum hydroxide [41].

The increase of friction with hardness in sliding machine element materials was discussed in [73] suggesting the importance of the adhesive component. It is also possible that the harder (and also more brittle) the material is, the more effectively it acts as an abrasive element between the material and the actual abradent after being removed from the nano-composite.

The general improvement in the wear resistance along with nanofiller content may be explained by the MWCNTs playing the role of spacers preventing a close contact between the abrader and the composite surface [74]. In such a case, the MWCNTs would be deformed into a graphene-like lamella, which would decrease the wear loss and the friction coefficient, as shown in [29]. On the contrary, the increase of the average mass loss (for MWCNTs 0.5 wt.% to 0.75 wt.%) could be a result of increased hardness and brittleness B.

Thermal conductivity could also be taken into consideration. As it is known, the influence of the filler on thermal conductivity is very significant [10]. It is possible that a higher thermal conductivity value results in a lower temperature in a sliding contact and therefore improved wear resistance for MWCNTs 2.0 wt.%. 

### 4.6. Surface Imaging

Bright field optical microscopy showed an uneven distribution of MWCNTs and agglomerations in a microscale (Figure 11). As a consequence of the uneven distribution for smaller mass concentrations of nanofiller, bright areas are visible where light shines from under the thin sample.

The pictures from the dark field microscopy and SEM confirm a large amount of detached material in a form of powder on the abrasive surface of the sample (Figure 12). 

Before the abrasion test, the characteristic surface obtained due to milling is visible. During the test, the detached material is spread over the surface of the sample, as a result of which the microscopic lines of white powder are visible in the photo from the optical microscope. This powder penetrates surface irregularities. As a result, surface irregularities become less visible in the SEM image compared to the situation before the test. The SEM images show that, according to classification of wear presented in [75], the observed mechanism is mainly adhesive wear.

## 5. Conclusions

HR-TEM pictures and HR-TEM EDX proved the properties of MWCNTs that occurred to be even better than those declared by the manufacturer. In the manufacturer’s specification, the declared purity of coal is 90 wt.%, while the test showed more than 95 wt.%.

Raman spectra, particularly for the highest MWCNT concentration show that carbon nanotubes were not damaged by sonication during the preparation of samples, and that they are present in the nanocomposite samples for all the applied mass fractions.

Optical microscope pictures showed microscopic agglomerations and TEM pictures showed carbon nanotube bundles proving the limitations of dispersivity method based on sonication only.

The wear resistance occurred to be in correlation with ultimate tensile strength. This may mean that the higher the tensile strength, the more difficult it is to tear the material away from the sample causing the mass loss. Moreover, together with tensile strength, maximal strain may influence the wear behavior making the material more deformable in the abrasion area, adapting to the rubbing abradant.

On the other hand, the mass loss and hardness occurred to be in correlation one with the other. For the low mass fraction of MWCNTs (0.25–0.5 wt.%), the mass loss and hardness decreased in comparison to neat resin, while for the higher weight percentage (0.75–1.0 wt.%) an increase in both values was noticed. This may indicate that the lower hardness of the material, the more it tends to form a carbon film covering the contact surfaces and act as a solid lubricant (as presented in [43]).

The study shows that in case of MWCNT-doped epoxy resin within the mass fraction of the nanofiller below 2.0 wt.%, the best wear resistance is achieved for the mass percentage between 0.25 and 0.5. For this reason, such proportions can be promising for structural elements of machines (e.g., aircraft).

## Figures and Tables

**Figure 1 materials-13-02696-f001:**
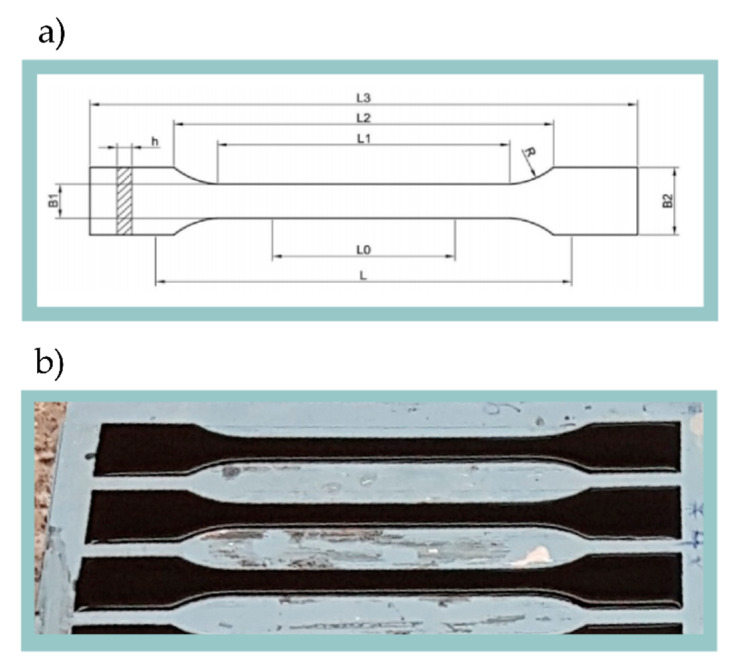
(**a**) Geometry of the sample for tensile test where: L3—Total length: 150 mm; L1—Length of parallel edges narrow zone: 60 mm; R—Radius: 60 mm; B2—Width at ends: 20 mm; B1—Width at narrow zone: 10 mm; h—Thickness: 4 mm; L0—Reference length: 50 mm; L—Length between clamps: 115 mm. (**b**) Nanocomposite casted to the silicone forms.

**Figure 2 materials-13-02696-f002:**
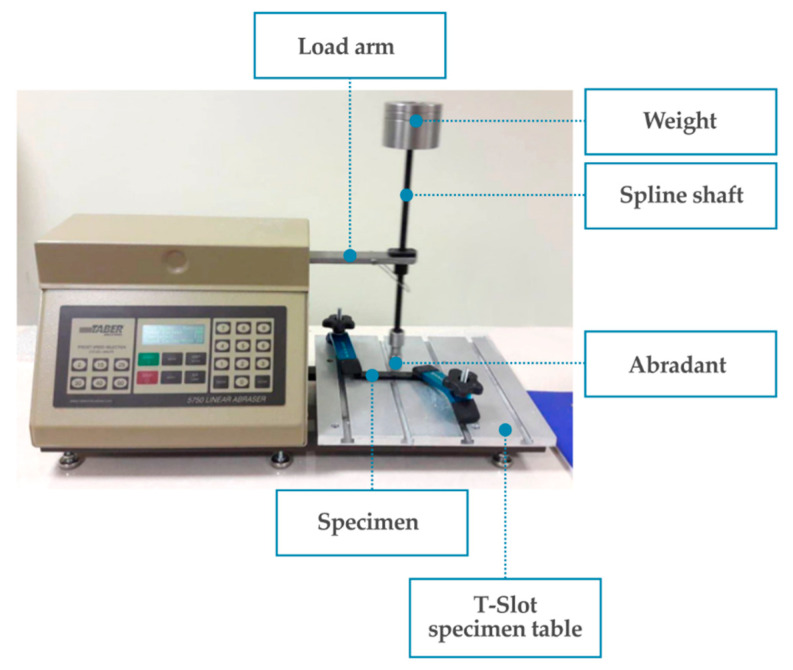
TABER linear abraser model 5750 during the abrasion test.

**Figure 3 materials-13-02696-f003:**
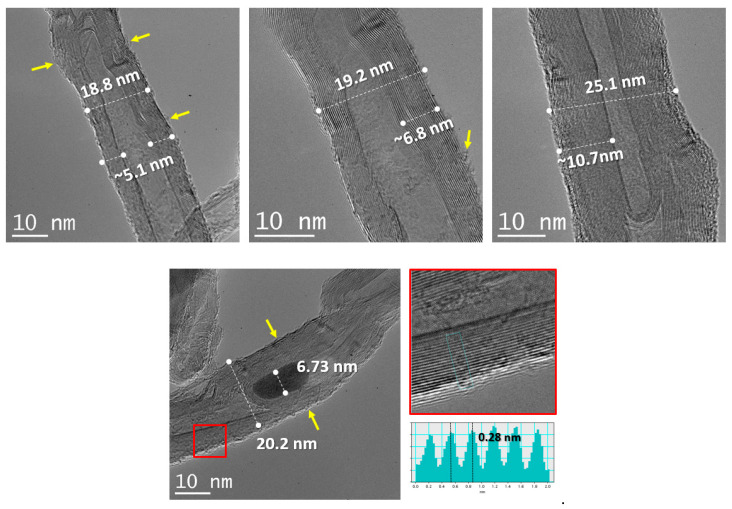
HR-TEM pictures of carbon nanotubes used for MWCNT/epoxy composite.

**Figure 4 materials-13-02696-f004:**
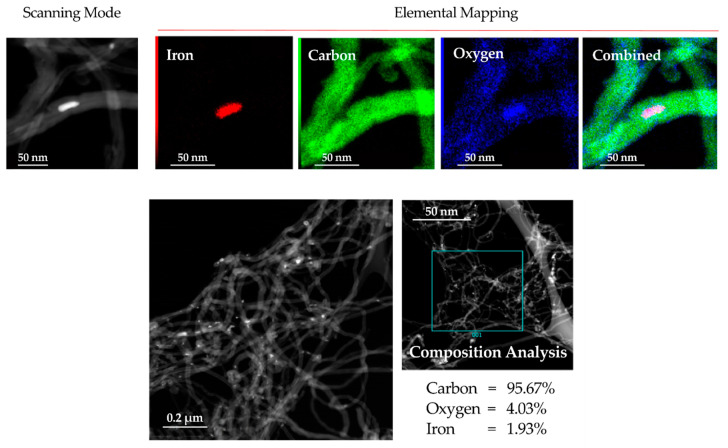
HR-TEM—EDX results of carbon nanotubes used for MWCNT/epoxy composite.

**Figure 5 materials-13-02696-f005:**
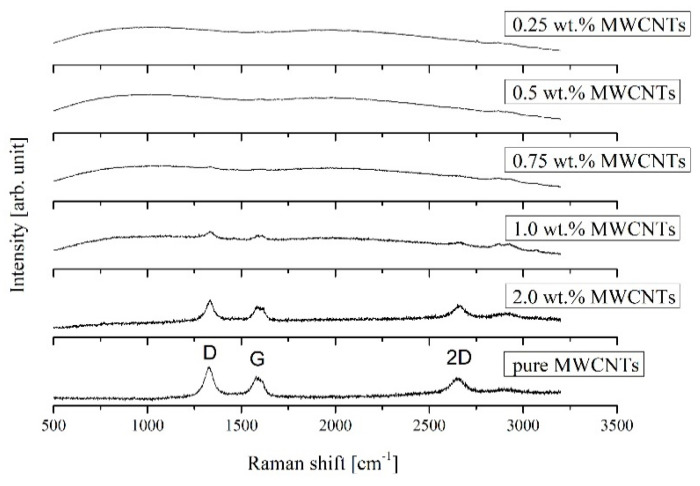
Raman shift for MWCNTs/epoxy composite samples with 0.25, 0.5, 0.75, 1.0, and 2.0 wt.% mass fractions of carbon nanotubes and pure MWCNTs for reference.

**Figure 6 materials-13-02696-f006:**
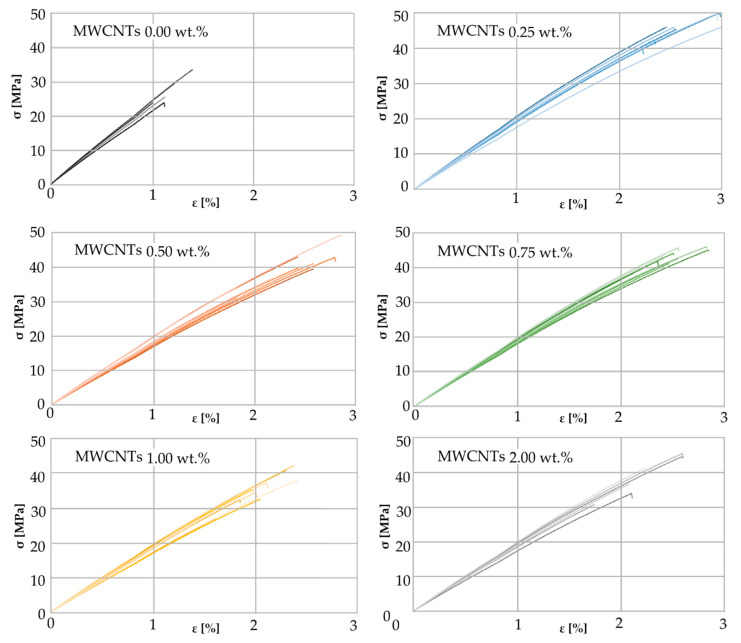
Stress–strain curves for different MWCNTs mass fractions.

**Figure 7 materials-13-02696-f007:**
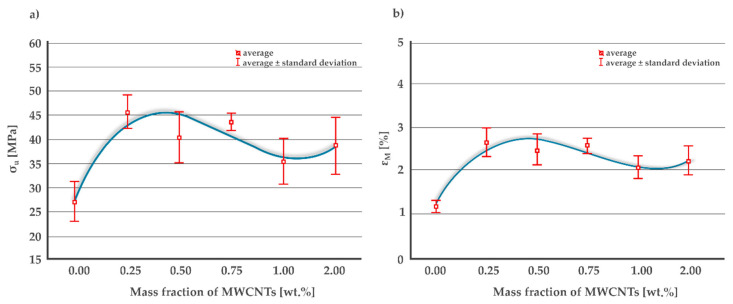
(**a**) Ultimate tensile strength and (**b**) nominal strain at break of the MWCNTs/epoxy composite for different MWCNTs mass fractions.

**Figure 8 materials-13-02696-f008:**
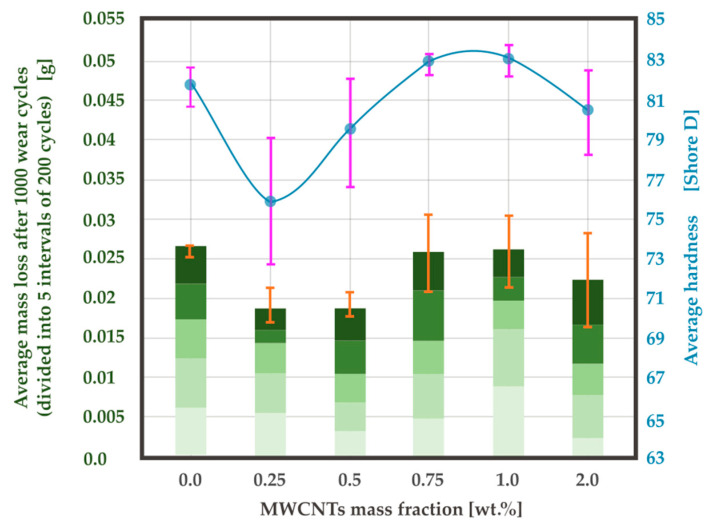
Correlation of mass loss and hardness in relation to MWCNTs mass fraction.

**Figure 9 materials-13-02696-f009:**
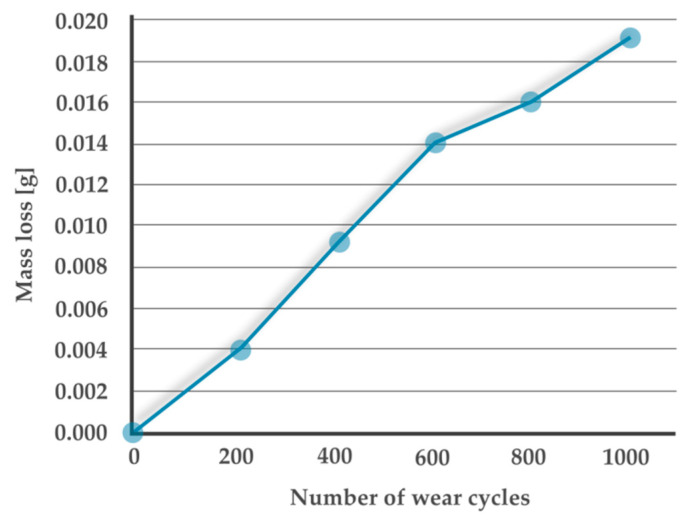
Wear process presented on the example of a MWCNTs 0.25 wt.% sample.

**Figure 10 materials-13-02696-f010:**
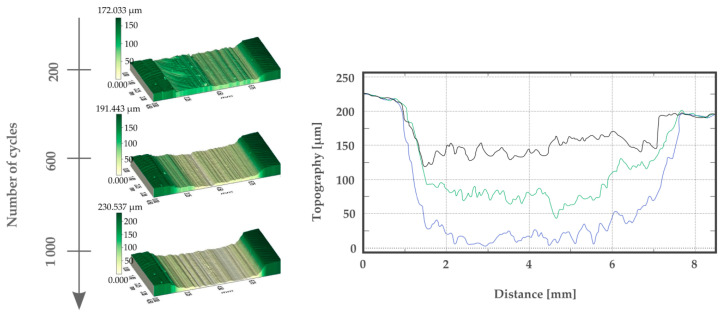
Topography and profiles of MWCNTs 0.25 wt.% sample after 200, 600, and 1000 wear cycles.

**Figure 11 materials-13-02696-f011:**
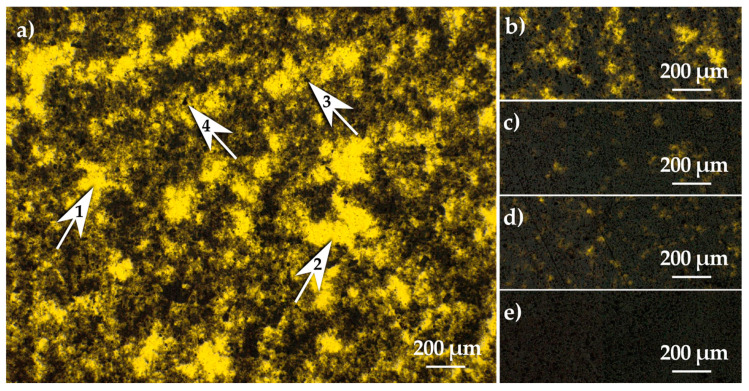
Bright field optical microscopy of MWCNTs (**a**) 0.25, (**b**) 0.5, (**c**) 0.75, (**d**) 0.1, and (**e**) 0.2 wt.%.

**Figure 12 materials-13-02696-f012:**
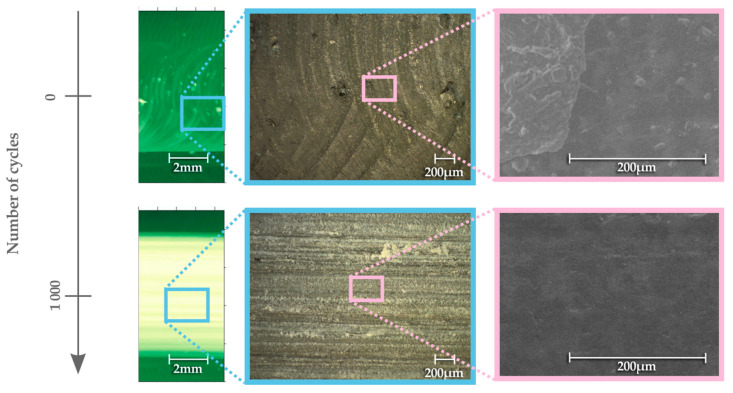
Profilometer topography, optical microscope images and SEM images of MWCNTs 0.25 wt.% sample after 0 and 1000 cycles of abrasive wear.

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
