# Peer review of "Effect of MWCNTs on Wear Behavior of Epoxy Resin for Aircraft Applications"

_materials, 2020, doi:10.3390/ma13122696_

Round 1
Reviewer 1 Report
The paper entitled “Effect of MWCNTs on wear behaviour of epoxy resin for aircraft applications” reports the study of mechanical properties of composites with CNTs. In particular, this article focus on the wear behaviour. The argument is not so originally but it could represent a particular case study in the sector.
I suggest the publication after minor revision here reported.
- Use always the indication of percentage in wt.% (see abstract)
- In samples preparation when you write. “After that, the composition of L285 and MWCNTs was mixed with H287 in 100:40 resin to hardener weight ratio as recommended by manufacturer.” Please specify how do you mix components. Do you have presence of air bubbles? How do you remove them?
- Cause and Effect Model (Fig. 1) seems to be not well defined. I suggest to add some explanations or remove it. In all figures, please, use the same character (Times New Roman).
- Raman spectroscopy section: please pay attention to the character corrections ( mMeasurement)
- Figure 5: scales in element mapping are not clear
- Figure 6: not really clear to read. I suggest to put in the same figures all spectra, normalized on D peak (as you done) but with a space among them. Could be interested to add also pure epoxy resin spectra?
- I suggest to add a study of MWCNTs dispersion and MWCNTs bundle presence. For example, using FESEM or TEM images.
Reviewer 2 Report
1. The introduction is too general and should be more focused on the wear properties of the epoxies filled with MWCNTs to reveal the state-of-the-art. Also the novelty and the aim of this study should be emphasized not in one sentence as it was done in the paper but in details.
2. The chosen filler fraction for the epoxy (0-2 wt.%) should be argued. Was it based on ref. [53] which is just modelling results? Moreover, there is no data on electrical properties and electrical percolation in the current paper. Therefore, this part of the introduction should be revised to focus more on the wear and mechanical properties which are discussed in the paper.
3. It was mentioned that optical microscope Olympus BX53M was applied to investigate agglomerates of MWCNTs but no micrographs and/or discussion were provided. In fact, the dispersion of MWCNTs should be analyzed not only at nano-scale (as it was done in Figure 4) but also show representative separation of the MWCNT at microscale since the nanoparticles may arrange to agglomarates and as a result transform to microparticles. Therefore, optical micrographs and discussion of the dispersion of MWCNTs should be added to the paper.
4. Representative stress-strain curves for tensile tests are the most informative for the analysis of the nanofiller effects. They should be added and discussed before Figure 7. Moreover, elastic modulus is also very important characteristic which couldn't be omitted when speaking about tensile properties of the materials. Lastly, the increase in maximal deformation due to increase of the nanofiller content is not in line with the literature where it opposite effect (decrease) is usually discussed. Therefore, detailed discussion with the explanation about such result is necessary together with providing references from the literature.
5. The sequence of Figures is incorrect and should be corrected. Figure 10 is located before Figure 8 in the text.
Taking into account these remarks it is recommended to revise the manuscript by doing major review.
Reviewer 3 Report
This work describes the modification of the epoxy system by carbon nanotubes in order to increase the physicomechanical properties. A huge number of similar works exist. The present work is done accurately at the experimental level and its quality is beyond doubt. However, from my point of view, it does not present fundamentally new results and is not suitable for publication in the journal Materials on the criterion of "scientific novelty." In addition, the disclosure of the structure-properties relationships in the work is minimal, the paper is more like a report than a scientific work. I believe that the article should be fundamentally revised before publication in a scientific journal. In addition, the following points need to be addressed in the future.
1) In section 2.1. Please indicate the chemical formula of resin L285 and hardener, if possible. Indicate what specific standards the resin meets and why it is important for this study.
2) Used nanotubes contain oxygen. What functional groups are present on the surface?
3) How can the reader verify that the dispersion of the nanotubes in the epoxy system was good enough? For dispersion, sonication for 10 minutes was used. Why is this time chosen?
4) As a rule, there is no need to provide photographs of devices for testing.
5) The authors observe the phenomenon of an increase in some properties of the material, but practically do not say anything about its causes (especially with the loss of the fact that the dependence is rather complex). The authors say that the improvement in properties is due to adhesion between the nanofiller and the matrix, but it is widely known that unmodified nanotubes are poorly compatible with the epoxy matrix. The term "more accurate crosslinking" also requires detailed explanations.
Round 2
Reviewer 2 Report
1. No efforts were done by the authors to improve the introduction as it was recommended previously. I still think that it is an important part of the paper and since the paper is focused on wear properties the introduction should include the state-of-the-art for these properties of the epoxies filled with MWCNTs. Also no changes were done to include the novelty and to emphasize the aim of this study in details.
2. The discussion provided by the authors regarding the chosen filler fraction for the epoxy (0-2 wt.%) should be added to the paper in short. No changes were done for the new submission.
3. The representative stress-strain curves were added by the authors but the discussion of them is too general. Also it is not clear what is on the curves. The legend is absent.
4. The results for the elastic modulus as a function of filler fraction should be compared with literature data.
Taking into account above remarks the paper is recommended for reconsidering after major review and including all emphasized points in the text.
Reviewer 3 Report
The authors improved the quality of the manuscript and excluded controversial points from the discussion. I still think that the novelty of this paper is low but the paper can be published in Materials because of valuable applied results and as a useful case study.
Round 3
Reviewer 2 Report
The authors made many corrections.
The only critical point left is stress-strain curves which should be provided for different filler fractions to show the filler effect.
On the one hand, the authors wrote that:"Stress-strain curves for all carbon nanotube mass fractions were similar, so the results for 0.25 wt.% samples were presented in Figure 6 as an example."
But on the other hand, based on further different results for tensile strength, strain at break and Young’s modulus the curves should reflect this nanofiller effect.
Therefore, it is suggested to add stress-strain curves for every filler content and qualitatively discuss the results for mechanical properties.
